# The Perceived Influence of Food and Beverage Posts on Social Media during the COVID-19 Pandemic: An Exploratory Study with U.S. Adolescents and Their Parents

Adam J. Kucharczuk * and Tracy L. Oliver

M. Louise Fitzpatrick College of Nursing, Villanova University, Villanova, PA 19085, USA
* Correspondence: akucharc@villanova.edu

**Abstract:** Additional time spent on social media (SM) due to nationwide lockdowns associated with the COVID-19 pandemic has increased adolescents' exposure to food and beverage (FB) advertisements, which may increase one's risk of developing unfavorable health outcomes. This study aimed to explore U.S. adolescents' and their parents' perceptions of social media's influence on adolescents' food and beverage preferences during the COVID-19 pandemic. Semi-structured focus groups were conducted virtually with seven dyads of sixth grade students and their parents (*n* = 14). Six themes were identified from the focus groups: (1) perceived increased accessibility to SM usage, (2) factors that increased consumption, (3) perceived increased recall of memorable aspects of FB advertisements, (4) parental observations of adolescents' less healthy eating behaviors, (5) parental influence over FB purchases, and (6) perceived increased engagement with food trends from SM. Increased SM use influenced adolescents' preference toward specific FB brands and possibly influenced consumption habits during the pandemic. Parents may be aware of the targeted marketing used on SM and may minimize some of this influence. Additionally, these findings should encourage parents and adolescent healthcare professionals to proactively discuss the marketing tactics FB companies use and continue to educate adolescents on the importance of maintaining healthy eating behaviors.

**Keywords:** social media; adolescents; food and beverage preferences; COVID-19; qualitative

## 1. Introduction

The use of social media (SM) has grown exponentially across generations within the past decade. It is estimated that approximately 93–95% of U.S. adolescents have at least one SM account, with 45% reporting they are online "almost constantly" and most having two or more accounts [1,2]. This high rate of SM use among adolescents varies around the world and ranges from approximately 17–50% in Swiss and Italian adolescents, respectively [3]. Current research suggests that the increase in the prevalence of SM use has adverse effects on adolescents' lives, e.g., on their mental health, eating habits, food preference concerns, and a proposed addictive nature towards SM use [4,5].

Mirroring adolescents' increased presence on SM, international food and beverage companies are expected to spend approximately $455.3 billion in 2022 to advertise on SM, a 20.4% increase from 2021 [6,7]. Companies that market unhealthy options such as PepsiCo, Coca-Cola, and Frito-Lay run hundreds of SM accounts with the intent to establish brand name preferences among adolescents [6]. Research suggests that adolescents are more likely to succumb to brands' marketing tactics when companies incorporate celebrities and influencers into the advertisement because of the appeal and adolescent desire to be like these individuals [3,5]. With this, the rise of influencers on SM is a new marketing strategy that is effective due to their extensive following, trust in the influencer, and covert product placement in SM posts [8,9]. Marketing through the use of SM influencers, such as product placement in SM posts on YouTube and mock Instagram profiles, has been found to increase 9–11-year-old children's unhealthy food intake after viewing an influencer with unhealthy

snacks [8,9]. When coupled with the increased SM use among adolescents, these tactics only put adolescents at more risk for viewing unhealthy food and beverage digital marketing [10].

The COVID-19 pandemic is increasing users' reliance and engagement with the internet and its content as a result of state-mandated restrictions and nationwide lockdowns. With adolescents staying home because of these efforts, their engagement in physical activities decreased, whereas their time on SM increased [11,12]. Seventy-two percent of surveyed parents were concerned that overusing SM during the pandemic has impacted their child's physical and mental health [11]. Consequently, the pandemic has inherently caused decreased physical activity, increased stress eating, bored consumption, and overconsumption of unhealthy foods, which may increase negative health risks such as obesity [12–14]. Furthermore, obesity puts an individual at risk for other lifelong health problems such as hypertension, type 2 diabetes, and pulmonary problems [15]. Previous studies have identified changes in consumption patterns during the COVID-19 pandemic. One study found Polish adults ate and snacked more during the pandemic, and Italian adults had an increased appetite that consequently led to an increase in junk food consumption, which potentially explains how 48.6% of the population gained weight during the pandemic [16,17]. When looking at consumption habit changes in children and adolescents, a Brazilian study found children and adolescents increased their consumption of soft drinks and decreased their consumption of healthy food options during the pandemic [18]. This study also found that the participants ate higher energy-dense foods that were high in fat and sugar while watching television [18]. Though these studies identify some of the consumption habit changes that occurred during the COVID-19 pandemic, most do not focus on adolescents or the potential influences of consumption, such as SM. Because adolescents are spending more time on SM during the pandemic, their risk for food and beverage advertisement exposure may increase and potentially have a greater influence on their food preferences than it did prior to the pandemic [11].

As primary food purchasers for the family, parents play an important role in regulating what food and beverage products are bought for the family. The pandemic evoked new stressors on families, which often led to less healthy feeding patterns and consumption habits for families, including more lenient food consumption rules, increased use of food to soothe, and unhealthy snacking [19,20]. Adolescents are often successful at influencing their parents' food choices and purchases by negotiating, persuading, and making demands to align with their food preferences [21–23]. The concern becomes when adolescents influence their parents' food choices and purchases based on less healthy food and beverage products they see advertised on SM.

SM has been found to have a lasting impact and influence on many facets of adolescents' lives, including mental health and academics. Specifically, food companies target adolescents through SM and may directly influence adolescents' eating behaviors by potentially producing a lasting food preference towards their products [5]. However, it remains largely unknown whether adolescents' increased time and exposure to both SM and targeted food marketing associated with the COVID-19 pandemic has caused an even greater impact on their food preferences. Previous studies that focused on SM use and food preference have examined behavioral changes during a time in which there was not rapid increases in SM use such as during COVID-19 [8,9,24]. Additionally, previous studies utilized mock SM feeds [9]. Mock feeds would not be able to incorporate the targeted marketing strategies participants experience in real life, producing a less-than-realistic simulation for the study. Thus, the objective of this study was to explore U.S. adolescents', aged 12–13, and their parents' perceptions of social media's influence on adolescents' food and beverage preferences during the COVID-19 pandemic using the participants' real-life experiences.

## 2. Materials and Methods

### 2.1. Research Design and Recruitment

This qualitative research design utilized group interviews to examine SM's influence on adolescents' food and beverage preferences during the COVID-19 pandemic and explore

their parents' perceptions of this phenomenon. A total of six group interviews were conducted virtually using Zoom. Three group interviews were held with the adolescents, and three group interviews were held for their parents. The group interviews were broken up into groups of 3, 3, 3, 3, 1, and 1. The last two focus groups became interviews with just one participant each due to anticipated participants not logging into the Zoom meeting. Each group interview lasted approximately one hour and was digitally recorded. The sample was recruited through a public school district located in northeastern Pennsylvania, USA. The study announcement was emailed to all sixth grade students by their principals as well as posted on the school district's SM accounts. Interested participants contacted the first author, who then provided more specific information regarding the study by sending the consent and assent forms. Sixth grade students were recruited because of their recent transition into middle school as well as emergence into adolescence. The study sought to recruit 40 participants through traditional email and SM recruitment efforts. The study also extended recruitment efforts by three months, and snowball sampling was added to the recruitment strategy to encourage enrollment through previous participants' contacts. However, the study was only successful in recruiting 14 participants ($n = 14$) after these additional recruitment efforts had been exhausted. The study was reviewed and approved by Villanova University's Institutional Review Board IRB-FY2021-214.

## 2.2. Study Design

The group interview sessions were held virtually using Zoom in July and September 2021. Days prior to each group interview session, verbal consent from the parents, parental consent for their adolescent, and, finally, verbal assent from the adolescent were collected. Following the consent and assent process, all participants were sent a demographics survey using Qualtrics. Semi-structured interview guides containing open-ended questions were developed by a multidisciplinary research team, including a dietitian and nurse, and were piloted with parents and adolescents of similar demographics. The interview guides were modified based on feedback and then used to guide the discussion. The research team explored adolescents' SM habits, food and beverage consumption patterns, and characteristics of food and beverage advertisements viewed on SM. Additionally, adolescents were prompted to consider possible food and beverage consumption influences by SM and changes that may have occurred during the COVID-19 pandemic lockdown. Sample group interview questions can be found in Tables 1 and 2.

**Table 1.** Example Questions from Semi-structured Group Interview Guide for Adolescents.

1. When and where do you usually use social media?

2. Can you describe how your social media use changed since the start of the COVID-19 pandemic lockdown?

3. Tell us about some of the food and beverage advertisements you see on social media.

4. How have your eating habits changed since the start of the COVID-19 pandemic lockdown?

5. How do you communicate to your parent/guardian about which food or beverages you'd like to eat or have them purchase?

6. Do you feel like you are directly influenced by the food and beverage advertisements you see on social media?

7. How do you think this has changed since the start of the COVID-19 pandemic lockdown?

**Table 2.** Example Questions from Semi-structured Group Interview Guide for Parents.

1. Do you feel like your sixth grade student's social media habits have changed since the start of the COVID-19 pandemic lockdown? How so?

2. Tell us about some of the food and beverage advertisements your sixth grade student has seen on social media.

3. How do your sixth grade students express their food preferences?

4. Tell us about any changes you have observed related to your sixth grade student's food and beverage preferences since the start of the pandemic and the lockdown.

5. Do you feel your sixth grade student is directly influenced by the food and beverage advertisements they see on social media?

6. How do you think this has changed since the start of the COVID-19 pandemic lockdown?

*2.3. Data Collection*

Both authors were present at each group interview session. The first author served as the lead moderator, and the second author took notes, managed the Zoom meeting, and sought clarifications by asking additional questions. The group interview sessions were audio recorded and transcribed using Zoom functions. The transcripts were validated by comparing the recorded audio with the transcripts for accuracy, and corrections were made.

*2.4. Data Analysis*

Descriptive statistics gathered from the demographics survey were used to describe the sample. Thematic analysis was applied to the group interview transcripts, as this technique is best suited for describing perceptions and experiences in qualitative studies [25]. This lead author independently developed a preliminary coding framework and codebook based on the transcriptions. This process was conducted separately for each set of group interview transcripts (adolescents vs. parents). Next, both authors independently coded the data using the qualitative software ATLAS.ti and the codebook that linked themes to defined codes [26]. Both authors took notes to describe possible themes among groups, and intercoder reliability was enhanced by comparing the coded text across participants and discussing discrepancies. Next, the authors met to examine the emergent themes. Finally, another round of analysis was conducted using the constant comparative method to refine themes by systematically comparing all relevant data until each aspect was discussed and categorized by both authors using the emergent themes [27].

**3. Results**

In total, seven adolescent–parent pairs who mirror the school district's demographics participated in six virtual group interviews. Six out of seven adolescents were aged 12 years old, with the seventh student being 13 years old. Four of the adolescents were male. Adolescents reported using SM an average of 1 h daily before the pandemic and 2.86 h daily during the pandemic, with an average daily increase of 1.86 h. This usage increase was largely a result of parents loosening pre-pandemic SM limits or restrictions. The SM platforms adolescents most frequently used were YouTube, Tik Tok, and Instagram, respectively. Five of the parents were aged 35–44 years, with the remaining two parents being aged 45–54 years. All parent participants were female and were married or in a domestic relationship. In addition, six parents reported changing rules for their adolescents on SM once the pandemic started. Additional demographic data can be found in Table 3, and SM usage data can be found in Table 4. Data reported in the preliminary surveys aligned with many participants' responses during the group interviews.

**Table 3.** Sociodemographic Characteristics of the Adolescent and Parent Participants.

| Characteristics | Adolescents (*n* = 7) | Parents (*n* = 7) |
|---|---|---|
| Age | | |
| 12 | 6 | |
| 13 | 1 | |
| 35–44 | | 5 |
| 45–54 | | 2 |
| Sex | | |
| Male | 4 | 0 |
| Female | 3 | 7 |
| Race | | |
| White or Caucasian | 5 | 5 |
| Asian | 1 | 1 |
| Other | 1 | 1 |
| Ethnicity | 1 | 1 |
| Hispanic, Latino, or Spanish | | |
| Marital Status | | |
| Married or in a domestic relationship | | 7 |
| Highest Degree or Completed School | | |
| Some college, or no degree | | 3 |
| Bachelor's degree (e.g., BA, BS) | | 2 |
| Master's degree (e.g., MA, MS, MEd) | | 1 |
| Doctorate or professional degree (e.g., MD, DDS, PhD) | | 1 |

**Table 4.** Adolescent Social Media Usage Characteristics Reported by the Adolescents and Their Parents.

| Characteristics | Adolescents (*n* = 7) | Parents (*n* = 7) |
|---|---|---|
| Average Social Media Usage | | |
| Pre-pandemic (h) | 1 | 1 |
| During Pandemic (h) | 2.86 | 2.86 |
| Average Net Change (h) | +1.86 | +1.71 |
| Self-Reported Social Media Platforms Used | | |
| Instagram | 2 | |
| Twitter | 1 | |
| Facebook | 1 | |
| Tik Tok | 3 | |
| YouTube | 7 | |
| Other * | 2 | |
| Social Media Rules | | |
| Duration Rules | 4 | 5 |
| Content Rules | 6 | 7 |
| Yes, rules changed since start of pandemic | 1 | 6 |
| No, rules did not change since start of pandemic | 4 | 1 |
| N/A, did not have rules before the pandemic | 2 | 0 |

* Other includes: Messenger, Discord.

After analyzing the group interview transcripts, three adolescent themes were identified: (1) perceived increased accessibility to SM usage, (2) factors that increased consumption, and (3) perceived increased recall of memorable aspects of food and beverage advertisements. Three parent themes were identified from the group interview analysis as well: (1) parental observations of adolescents' less healthy eating behaviors, (2) parental influence over food and beverage purchases, and (3) perceived increased engagement with food trends on SM. Compilations of the qualitative themes, their emerging themes from data analysis, and supporting quotes can be found in Tables 5 and 6.

**Table 5.** Adolescent Qualitative Themes (n = 7).

| Final Themes | Emerging Themes | Supporting Quotes |
|---|---|---|
| **Perceived increased accessibility to SM usage** | New to SM | *"I started using social media during the pandemic because there was not much else to do."* |
| | Increase in SM use frequency | *"I didn't use social media before the pandemic, but now I use it to talk to my friends since I can't see them."* |
| | Advertisement frequency | *"[I] definitely used [social media] more than I did in past school years because after the Zooms, I wouldn't have one for the next like hour or so, so I just go on [social media] a lot more often."* |
| | | *"I've definitely started to see more ads on social media since the pandemic started."* |
| | | *"It's weird because I'm not seeing more advertisements because I'm watching more, I've seen more advertisements because there are more. Like now there's like a way wider variety of ads than there were before the start of the pandemic."* |
| **Factors that increased consumption** | Bored consumption | *"Definitely boredom."* |
| | Influence to purchasing intent | *"My cousin has a YouTube [channel] and while they were here, they wanted to try [hot chocolate bombs] out [to post]."* |
| | Consumption habit | *"I buy them more because I know that [Gatorade] has an item that I need, which is electrolytes and electrolytes help me kick up my energy when I'm tired. If I'm playing baseball and I get tired, I can easily just drink a bit of Gatorade and charge up."* |
| **Perceived increased recall of memorable aspects of FB advertisements** | Advertisement characteristics | *Mr. Beast, Travis Scott, J. Balvin, Kevin Durant, Stephen Curry, Alex Morgan, and Shaq* |
| | Advertisement frequency | *"A lot of the people that I follow on social media promote other people and their [brands]."* |
| | Advertisement influence | *McDonalds, Burger King, Grub Hub, Starbucks, Lays, Doritos, Beast Burger, Bang, Takis, Gatorade* |
| | | *"There'd be like Starbucks ads, and I'd be like, 'Oh, I want to try this drink!'"* |

**Table 6.** Adolescent Qualitative Themes (n = 7).

| Final Themes | Emerging Themes | Supporting Quotes |
|---|---|---|
| **Parental observations of adolescents' less healthy eating behaviors** | Bored consumption | *"Normally wouldn't eat in the morning and then he was suddenly eating snacks in the morning, like after breakfast."* |
| | Consumption habit | *"He really was like every day asking for candy."* |
| | Unhealthy product | *"Every day he almost asked me, 'Can I eat some ice cream?' maybe right after lunch, or any time or snacks."* |
| | | *"Oh my gosh, like they just kept [eating]."* |
| | | *"She's kind of been left to handle the household by herself and left [with] her own devices, and lots of screen time, lots of snacking, lots of boredom . . . Sometime I would catch her on the couch with the entire bag of chips."* |
| | | *"I also feel like because he has verbalized to me that he will bored eat . . . he just snack, snack, snacks. So wanting more of the unhealthy snack food, and then consuming more of it has happened."* |
| | | *"Unhealthy [ads], unhealthy by far."* |
| **Parental influence over FB purchases** | Product price | *"It was like 13 bucks for a bag of 10 [jelly fruit] or something ridiculous . . . so there was no way she was getting that."* |
| | Unhealthy product | *"We hardly eat those foods."* |
| | | *"Total sugar water."* |
| | | *"Never [goes] to the grocery store with me."* |

**Table 6.** *Cont.*

| Final Themes | Emerging Themes | Supporting Quotes |
|---|---|---|
| **Perceived increased engagement with food trends seen on SM** | Fads from SM<br><br>Influencers<br>Advertisement influence | *"The hot chocolate bombs were all rage, so all they wanted was to drink hot chocolate bombs all day and to find hot chocolate bombs."*<br>*"We made our own."*<br>*"Did this recipe about some pasta that came out on Tik Tok."*<br>*"He tried to tell me, 'Let's do that noodle challenge!'"*<br>*"The one beautiful thing that came out of [the pandemic] is my kids started cooking."* |

*3.1. Adolescent Theme 1: Perceived Increase Accessibility to Social Media Usage*

Adolescent participants noted their increase in SM usage since the start of the pandemic. Some adolescents explained that they began using SM for the first time during the pandemic or used SM more frequently and for longer durations, exposing them to more food and beverage advertisements on SM. Adolescent 3 stated, "*I started using social media during the pandemic because there was not much else to do.*" Adolescent 5 shared a similar experience: "*I didn't use social media before the pandemic, but now I use it to talk to my friends since I can't see them.*" The adolescent participants also indicated their increased SM usage might have stemmed from having less schoolwork and activities, leading to more free time at home. Adolescent 1 explained, "*[I] definitely used [social media] more than I did in past school years because after the Zooms, I wouldn't have one for the next like hour or so, so I just go on [social media] a lot more often.*" When asked about the frequency of SM advertisements they were seeing, Adolescent 3 shared, "*I've definitely started to see more ads on social media since the pandemic started.*" Adolescent participants clarified that they saw more advertisements not because they used SM more, but because they were more frequent in quantity. Adolescent 6 explained, "*It's weird because I'm not seeing more advertisements because I'm watching more, I've seen more advertisements because there are more. Like now there's like a way wider variety of ads than there were before the start of the pandemic.*" Adolescents 1, 2, 4, 5, and 6 also agreed to seeing more food and beverage advertisements during the pandemic.

*3.2. Adolescent Theme 2: Factors That Increased Consumption*

The adolescents described instances that led to an increase in their food and beverage consumption after that food or beverage or similar food or beverage was seen on SM. Boredom and a general misconception about the benefits of food and beverage products seen on SM were noted to be drivers that influenced consumption. When asked what prompts them to snack while on SM, Adolescent 1 stated, "*Definitely boredom.*" Though the adolescents may not have reported consuming the same products that were seen on SM because they were not available in the home, they did consume similar food and beverages that were available while bored. For example, adolescents reported seeing Lays and Doritos on SM; however, they may have eaten other brands of chips that were available when they experienced boredom. Three parents also expressed that their adolescent ate out of boredom and described how the shutdown of the pandemic increased their adolescents' boredom from lack of engagement in outside activities. Another adolescent shared potential factors that increased their food consumption of a particular food when he described trying food trends from SM and then posting an online review about it. He shared, "*My cousin has a YouTube [channel] and while they were here, they wanted to try [hot chocolate bombs] out [to post].*" Not only had they originally seen the food in SM posts, but they then consumed that food item and posted it online to share with others. Adolescents also shared instances of consuming products based on the products' perceived nutritional benefits; however, oftentimes, these beliefs were based on successful marketing misconceptions about nutrition. Some of these products included Gatorade and Muscle Milk, which are often marketed to consumers as a product that will not only provide essential nutrients but may also aide in athletic performance. Adolescent 7 described seeing several Gatorade advertisements that featured baseball players and gymnasts. He explained, "*I buy them*

*more because I know that [Gatorade] has an item that I need, which is electrolytes and electrolytes help me kick up my energy when I'm tired. If I'm playing baseball and I get tired, I can easily just drink a bit of Gatorade and charge up.*" Marketing a product's misconceived nutrition benefits may be enough to influence consumption, but the influence may be far greater when coupled with the endorsement of a successful athlete.

### 3.3. Adolescent Theme 3: Perceived Increased Recall of Memorable Aspects of Food and Beverage Advertisements

Adolescents recalled various aspects such as specific people, brands, and types of foods that were incorporated into food and beverage advertisements and seen during the pandemic. Adolescents recalled influencers, YouTubers, celebrities, and athletes who were in advertisements. These people included Mr. Beast, Travis Scott, J. Balvin, Kevin Durant, Stephen Curry, Alex Morgan, and Shaq. Adolescent 1 shared, "*A lot of the people that I follow on social media promote other people and their [brands].*" Some of these celebrities and influencers were associated with McDonalds' burger specials, such as the Travis Scott burger, and Stephen Curry in Gatorade advertisements. Adolescents recalled the following food and beverage brands and online food ordering and delivery platforms: McDonalds, Burger King, Grub Hub, Starbucks, Lays, Doritos, Beast Burger, Bang, Takis, and Gatorade. Adolescents also identified some food and beverage products, which included soda, jelly fruit, and energy drinks. Adolescents reported wanting to try these products because of the advertisements' appeal. Adolescent 4 shared, "*There'd be like Starbucks ads, and I'd be like, 'Oh, I want to try this drink'*" after seeing this advertisement on YouTube. With increased recognition and recall of specific brands with their associated influencers, adolescents may desire to try the food and beverage products advertised or seen on SM.

### 3.4. Parent Theme 1: Parental Observations of Adolescents' Less Healthy Eating Behaviors

Parents noted less healthy eating behaviors among their adolescents since the start of the pandemic, some of which included consuming food and beverages more frequently and/or in greater amounts and snacking while on SM. Parents hypothesized their adolescents' increase in consumption could be attributed to boredom or loneliness. Parent 6 explained that her son "*normally wouldn't eat in the morning and then he was suddenly eating snacks in the morning, like after breakfast.*" She also added, "*He really was like every day asking for candy.*" Parent 5 shared a similar experience: "*Every day he almost asked me, 'Can I eat some ice cream?' maybe right after lunch, or any time or snacks.*" Similarly, Parent 4 was shocked at how much her adolescent was eating during the pandemic: "*Oh my gosh, like they just kept [eating].*" In agreement with their adolescents, parents also noted their adolescents were bored while eating and on SM. Parent 1 shared, "*She's kind of been left to handle the household by herself and left [with] her own devices, and lots of screen time, lots of snacking, lots of boredom . . . Sometime I would catch her on the couch with the entire bag of chips.*" Parent 3 stated, "*I also feel like because he has verbalized to me that he will bored eat . . . he just snack, snack, snacks. So wanting more of the unhealthy snack food, and then consuming more of it has happened.*" These responses depict a potential cycle of bored consumption while using SM and simultaneously being exposed to more food and beverage advertisements as described above. Parents believed this increase in less healthy eating occurred because the adolescents participated in fewer activities and stayed at home more than before the pandemic. When asked what kinds of advertisements their adolescents were seeing on SM, Parents 1 and 2 responded, "*Unhealthy [ads], unhealthy by far.*" Parents 1, 2, 3, and 7 believe these unhealthy advertisements are influencing their adolescents' food and beverage preferences.

### 3.5. Parent Theme 2: Parental Influence over Food and Beverage Purchases

Parent responses indicated that they, the parents, continue to have influence over the food and beverage products coming home. Parents considered food and beverage product prices, healthfulness of the product, and household needs before purchasing. Parents shared that the product's price impacts their decision of whether to purchase the food or

beverage product that the adolescent requests. Parent 2 explained her decision to decline her daughter's request for a food item that she saw on SM: "*It was like 13 bucks for a bag of 10 [jelly fruit] or something ridiculous . . . so there was no way she was getting that.*" Parents also consider how healthy a product is before purchasing. Parent 7 shared a time when her son asked for the Travis Scott Burger and J. Balvin Meal from McDonalds after seeing them on SM. She decided not to buy these items because "*We hardly eat those foods,*" insinuating these items were unhealthy. Parents reported not allowing their adolescents to have sugary products, such as Kool Aid Jammers, because they are "*Total sugar water.*" Parents also described not allowing their adolescents to purchase imported products that were marketed on SM out of concern for the products' unregulated ingredients. Even if her adolescent asks her for a specific food or beverage item, Parent 5 explained she will not always buy it, because her adolescent "*Never [goes] to the grocery store with me.*" This suggests that parents are often free from adolescent persuasion while shopping alone and implies that if the adolescent does grocery shop with the parent, parents may be more likely to give in to their adolescents' food and beverage purchase requests. Ultimately, parents make the final decision regarding what food and beverages are purchased, which is often based on the product's price and how healthy it is, but not always free from adolescent influence.

### 3.6. Parental Theme 3: Perceived Increased Engagement with Food Trends from Social Media

Parents discussed food trends that were seen on SM and their adolescent's engagement with the trend. Because activities used to entertain their adolescents were limited during the pandemic, parents described times in which food and beverage SM posts were used as a way to engage with their children. Parent 3 shared, "*The hot chocolate bombs were all rage, so all they wanted was to drink hot chocolate bombs all day and to find hot chocolate bombs.*" She later added that she believes her adolescent saw them on Tik Tok more than commercials, and they were not consumed prior to the pandemic. Additionally, Parent 1 explained that they could not find certain products that were requested by their adolescent, such as Jelly Fruit, so "*We made our own.*" This may suggest that parents also desired to try certain food items they saw on SM and went to great efforts to consume these items. Parents also shared that their adolescents asked to make trending recipes found on SM. Parent 7 explained that her adolescent "*Did this recipe about some pasta that came out on Tik Tok,*" and was eager to try it herself. She later described a time when her adolescent asked her to partake in a food challenge that he had seen on Tik Tok because there was not much else do to at home: "*He tried to tell me, 'Let's do that noodle challenge'*", but ultimately, they did not complete the challenge because the product was too spicy. Lastly, Parent 4 shared a similar and more positive cooking experience with her adolescent. She observed, "*The one beautiful thing that came out of [the pandemic] is my kids started cooking*", and they found recipes from their Amazon Echo Show and cousin's nutrition Instagram account. Therefore, in this case, SM provided a positive exposure to food and beverages. In addition, SM offered new alternatives for parents to engage with and entertain their adolescents while they were forced to stay home during the pandemic.

## 4. Discussion

The results from this study both support and expand upon current research regarding SM's potential influence on U.S. adolescents' food and beverage preferences. The COVID-19 pandemic presented a unique phenomenon that led to the development of six qualitative data themes after discussion with adolescent and parent participants.

Both parent and adolescent participants acknowledged a perceived increase in accessibility to SM usage since the start of the COVID-19 pandemic. The increase in SM use during the pandemic has also been noted in other recent reports [24,28,29]. With increased time spent on SM, adolescents were exposed to more advertisements than before the pandemic, corresponding with Forbes's report that U.S. companies spent 36% more on SM advertising than in 2020 [30,31]. Increased exposure to advertisements that incorporate celebrities or influencers may increase brand recall and recognition [5,32]. This may be supported when

adolescents brought up influencers like Mr. Beast, Stephen Curry, and Alex Morgan, all of whom were featured in advertisements for brands such as McDonalds, Starbucks, Takis, and Gatorade. Though most participants described viewing advertisements for unhealthy products on SM, it is important to note that some participants shared that they also saw healthier options on SM and consumed them [24].

Parents and adolescents also noticed differences in consumption habits since the start of the pandemic, such as bored consumption and consuming less healthy food and beverage options. In a similar study, Brazilian children and adolescents increased their consumption of soft drinks while also decreasing their consumption of healthy products during the pandemic [18]. Parents further reported seeing their adolescent eating more frequently and consuming unhealthy products when snacking, particularly while using SM. These findings support comparable studies that found adults also ate and snacked more during the pandemic, especially junk food [16,17]. Most of the food and beverages they consumed were either seen on SM or were similar products that were available in the home. This was also seen in a study examining the change in families' food-related practices following COVID-19 [33]. Vaughan and colleagues reported families' increased unhealthy snacking and grazing in response to the pandemic, which was also shared by participants [33]. Interestingly, two parents noted concern that their adolescents' BMIs increased at their latest health visit, moving them up on their growth curves and likely showing some of the immediate health concerns coming from the pandemic.

Although parents and adolescents described ways SM has influenced adolescents' food and beverage preferences, parents indicated that they still have control over food and beverage purchases being brought into the home. Several studies discussed parental influences on adolescents' purchases of unhealthy products, including monitoring and limiting the consumption of unhealthy products and effectively saying no to purchase requests [34–36]. However, it is important to consider that the adolescent participants in this study were on the lower end of the adolescent age range. As they become more independent, it is likely these parental influences will decline, enabling the adolescents to make their own decisions when purchasing food and beverage products based on their previously developed food and beverage preferences [37].

*4.1. Limitations*

This study's findings have limitations that should be considered during interpretation. The study had a small, nonrandom sample (*n* = 14) in one suburban school district in the U.S., challenging the comprehensiveness of participant responses. As mentioned in the introduction, adolescents around the globe use SM in varying amounts; therefore, the results presented in this study may not represent the same kind of influence in other countries [3]. Additionally, the small sample size may have limited participant responses and the qualitative data may not have achieved saturation. Since the study took place in early summer, there could have been decreased participation due to conflicts in summer scheduling as well as summer activities beginning. Although the study's participants had a relatively uniform demographic profile that mirrored the school district's population, the demographics of this study were not very diverse. For example, all parent participants reported being female and married. Additionally, given the voluntary nature and time commitment needed to participate in this study, interested individuals may not have had the opportunity to participate in the study and offer their unique perspectives. With other sociodemographic statuses being underrepresented, generalization to larger, more diverse populations, including other countries, should be performed with caution. Lastly, hesitancy towards sharing unique experiences in front of others might have limited some of the participants' responses, underreporting the influence of SM.

*4.2. Implications*

The current study provides preliminary support that suggests SM may have perceived influences on adolescents' food preferences, their parents' perceptions of this engagement,

and how the COVID-19 pandemic might have impacted these experiences. Adolescents' vulnerability to SM's marketing tactics presents a concern when brands are successful at marketing less healthy products and leads to a desire to try these products [4,5]. If frequent consumption of that less healthy product ensues, the adolescents may increase their risk of developing future health consequences, such as obesity, or complicating their current health conditions. Therefore, understanding factors that influence adolescents' food preferences, such as SM, is crucial when healthcare providers perform dietary assessments and provide dietary recommendations to adolescents and their families. In addition to assessing and making dietary recommendations, it is also important for healthcare providers to teach families about the lasting impact FB marketing on SM can have on adolescents. Though the COVID-19 adverse experiences have been mitigated, the food preferences and habits that developed during the pandemic could last into the long term, lending themselves to various health implications. With this, replication of this study internationally is recommended in order to understand how SM may influence adolescents' FB preferences differently in particular countries.

Public health policy and advertisement disclosure may be beneficial in reducing the impact of SM influence on adolescents. International policies regarding SM food and beverage advertisements vary greatly, but generally lack policies that restrict unhealthy food and beverage advertisements on SM platforms [38]. Policies should limit targeted advertisements to adolescents and require SM platforms to identify posts that contain marketed products so that viewers are aware of the advertisement. For example, in response to misinformation spread about the pandemic on SM, Instagram introduced an informational label on all posts that mentioned 'COVID-19', so users can click and read credible information [39]. The implementation of these restrictions may be beneficial in protecting adolescents from being inadvertently marketed to. SM use is ever evolving, including the emergence of livestreaming platforms, such as Twitch, during the pandemic. Twitch was recently found to expose users at high levels to food and beverage brands and products [31]. This signifies how brands introduce new marketing tactics to target consumers and the need for new interventions to be developed to protect adolescents from exposure. With effective public health policies in place, SM's influence on adolescents' food preferences can be minimized, hopefully limiting future potential health risks.

**Author Contributions:** Conceptualization, A.J.K. and T.L.O.; Data curation, A.J.K. and T.L.O.; Formal analysis, A.J.K. and T.L.O.; Funding acquisition, A.J.K. and T.L.O.; Investigation, A.J.K. and T.L.O.; Methodology, A.J.K. and T.L.O.; Project administration, A.J.K.; Resources, A.J.K.; Supervision, T.L.O.; Validation, A.J.K. and T.L.O.; Visualization, A.J.K.; Writing—original draft, A.J.K. and T.L.O.; Writing—review and editing, A.J.K. and T.L.O. All authors have read and agreed to the published version of the manuscript.

**Funding:** This study was supported by Villanova University's Undergraduate Research Fellowship Program (VURF).

**Institutional Review Board Statement:** This study was conducted according to the guidelines of the Declaration of Helsinki and approved by the Institutional Review Board of Villanova University (Protocol Number: IRB-FY2021-214; approved 29 June 2021, 3 September 2021).

**Informed Consent Statement:** Informed consent was obtained from all subjects involved in the study.

**Data Availability Statement:** The data presented in this study are available on request from the corresponding author. The data are not publicly available due to privacy and ethical restrictions.

**Acknowledgments:** The authors thank the school district administrators, Elizabeth B. Dowdell, Lauren DePiero, parent participants, and adolescent participants.

**Conflicts of Interest:** The authors declare no conflict of interest.

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
