# Peer review of "The Perceived Influence of Food and Beverage Posts on Social Media during the COVID-19 Pandemic: An Exploratory Study with U.S. Adolescents and Their Parents"

_adolescents, doi:10.3390/adolescents2030031_

Round 1
Reviewer 1 Report
Authors argue that "It is estimated that approximately 93-95% of adolescents have at least one 27 SM account, with 45% report being online" almost constantly "and most having two or 28 more accounts [1,2]." Which country do these statistics refer to?
The authors report that "Marketing through the use of SM influencers, like 41 product placement in SM posts, has been found to increase children’s unhealthy food in-42 take after viewing an influencer with unhealthy snacks [7,8]." It is not clear what contexts the cited studies refer to and which samples they considered. Please, provide more information so as to contextualize the results of the studies reported in the paper.
In the introduction, I believe that the limitations of the studies cited should be further emphasized, so as to make the relevance of the study objectives clearer. Furthermore, again in the introduction, there is no focus on the cultural context within which the authors carried out the study. Why is it relevant to study such links in that specific cultural context?
The number of participants is a relevant problem for the significance of the results obtained. I advise the authors to proceed with a calculation of the power of the effect (effect size).
Not enough information was provided on the study participants.
Please provide more specifics on how your participants were recruited, and what population does the sample represent? Why is that population important to your questions? How does your sample represent the population of interest? We all have to make decisions regarding limiting the sample. In what ways is your sample representative of the population of interest? Who is left out? Also, please provide additional information about the sample.
Please provide more specific information on how to participate. Why is that population important to your questions? How does your sample represent the population of interest? We all have to make decisions regarding limiting the sample. In what ways is your sample representative of the population of interest? Who is left out? Also, please provide additional information about the sample. For example, what racial diversity was present, if any? What percentage of the sample identified as sexual minorities? What was the nature of the university from which participants were recruited?
Although the results may be interesting, the study's limitations do not reveal their significance. Furthermore, the discussion turns out to be quite weak..
I believe it is very important to highlight the different applicative aspects of the present study and the possible practical implications. This allows the reader to understand the relevance of this topic and the importance of conducting subsequent studies in this area of ​​research.
Author Response
Thank you for the feedback and the opportunity to revise. Please see the attached document detailing the changes we have made.

Reviewer 2 Report
The manuscript adolescents-1836303 entitled "The Perceived Influence of Food and Beverage Posts on Social Media During the COVID-19 Pandemic: An Exploratory Study with Adolescents and their Parents" addresses the influence of consumption and content displayed on Social Media on adolescents' eating habits, using qualitative research. This methodological approach is adequate to meet the objective, as it is an exploratory study.
The manuscript is clear, brief, precise, relevant and generally presented in a structured way. The title captures the important information in the manuscript and is also accurate, including the term "exploratory study", the abstract presented is adequate and includes the relevant aspects and the objective is adequately stated. The results are presented as expected and are concise in each of the categories. The discussion is presented in an orderly manner.
However, it is suggested to improve the material and methods section, improving the level of concreteness and precision of some aspects.
For these reasons, a minor revision of the present version is recommended before publication.
Specific remarks:
1- Introduction:
The wording is concise and in the right direction, from the concrete to the particular.
It is suggested to improve the state of the question, including the age of the participants. Although this aspect is detailed in the results, it would be useful to refer to it in the introduction in order to situate the reader.
It is also suggested to include some reference to the change in adolescents' eating habits during the pandemic: is there an identified change in eating habits during the pandemic? if the answer is yes, it is appropriate to analyse the influence of social networks on the change. Although this aspect is addressed in L52 to L55, it is addressed only briefly. Going into it in more depth would considerably improve the starting point. I clarify this aspect further in the suggestions made in the discussion section.
2. Materials and Methods
2.1. Research Design and Recruitment
At some points in the manuscript reference is made to 6 adolescents and at others to 7. It is suggested to clarify this aspect and to improve the wording and increase the precision in how the interview groups are constituted, detailing the number and characteristics of the participants in each of them.
The title of sub-section 2.3. and 2.4. is the same. 2.3. Data Collection and 2.4. Data Collection. require clarification or modification,
3. Results: are detailed according to the established categories. The tables are timely
4. Discussion; the results are adequately discussed, although there is a lack of discussion with a similar study, if any, on the influence of social networks on the lifestyle habits of adolescents and on the pattern of consumption of other substances.
The information contained in L315 and L316, a recent study found that people who reported greater boredom during the pandemic increased their consumption of food and snacks [29]" refers more to the state of the question than to the discussion. In this section, the focus is on whether the results of that study are consistent with those reflected in this manuscript and the points of overlap or divergence between the two.
Author Response

(The authors gave the same response as above.)

Reviewer 3 Report
This qualitative study explores adolescents’ (12-13 years of age) and their parents’ perception of social media (SM) usage and its influence on food and beverage preferences since COVID-19 pandemic. Overall, the introductions very well justifies the need and importance of this research topic of social media usage and health (as related to food marketing and influences on food preferences), particularly since the COVID – 19 pandemic in this age group. However, methods sections needs some revision as described in the comments below.
Given, the qualitative nature of the study and small sample size (insufficient to reach data saturation), it would enhance the rigor of the study if the authors provided more information on analysis of qualitative data. A separate section titled “data analysis” describing in details the process of interview guide development (including the nature of the research team who developed it), coding, identify, refining and deciding on the final themes and categorizing them is much needed.
Introduction
Line 31: I believe the authors are referring to the addictive nature of social media usage not food and beverage consumption. Please clarify.
Methods:
Line 111 and 119: Both sub-headings read “Data Collection".
Give, the qualitative nature of the study, it would be informative to the readers if the authors provided more information on analysis of qualitative data including the nature of the research team. A separate section titled “data analysis” describing in details the process of coding, identify, refining and deciding on the final themes is needed.
The authors need to provide more information on the following:
Coding: did the first author have a start list of codes (e.g. from literature review) or were the codes generated independently just using the transcripts? What was the process of refining the codes as more data was analyzed, and how and at what point of data analysis was the coding structure finalized? What was the inter-coder reliability in the first step?
Line 130: The authors mention categories in this section. Please clarify if the categories were preconceived or developed based on the emergent themes. Who developed the categories (first or second author)? How these categories were developed?
Line 100: More details on the process of interview guide are required. For example, did any experts (such as child psychologist, nutritionist etc.) provide feedback to refine the interview guide? Who was included in the research team?
It would be beneficial if authors provided in a tabular form emerging themes from the “raw data”.
In addition, providing coding structure as supplementary material would be helpful.
Results:
Line 261: “insinuating that these items were not unhealthy” – did the authors mean “insinuating that these items were unhealthy”. The statement seems to contradict the quote provided.
Author Response

(The authors gave the same response as above.)
